# The Burden of Peritoneal Metastases from Gastric Cancer: A Systematic Review on the Incidence, Risk Factors and Survival

**DOI:** 10.3390/jcm10214882

**Published:** 2021-10-23

**Authors:** Anouk Rijken, Robin J. Lurvink, Misha D. P. Luyer, Grard A. P. Nieuwenhuijzen, Felice N. van Erning, Johanna W. van Sandick, Ignace H. J. T. de Hingh

**Affiliations:** 1Department of Surgery, Catharina Cancer Institute, Catharina Hospital, 5602 ZA Eindhoven, The Netherlands; anouk.rijken@catharinaziekenhuis.nl (A.R.); robin.lurvink@catharinaziekenhuis.nl (R.J.L.); misha.luyer@catharinaziekenhuis.nl (M.D.P.L.); grard.nieuwenhuijzen@catharinaziekenhuis.nl (G.A.P.N.); 2Department of Research and Development, Netherlands Comprehensive Cancer Organization, 3511 DT Utrecht, The Netherlands; f.vanerning@iknl.nl; 3Department of Surgery, Antoni van Leeuwenhoek—Netherlands Cancer Institute, 1066 CX Amsterdam, The Netherlands; j.v.sandick@nki.nl; 4GROW—School for Oncology and Development Biology, Maastricht University, 6229 ER Maastricht, The Netherlands

**Keywords:** peritoneal metastases, gastric cancer, incidence, treatment, survival

## Abstract

The peritoneum is a common metastatic site in gastric cancer. This systematic review provides an overview of the incidence, risk factors and survival of synchronous peritoneal metastases from gastric cancer. A systematic search was performed to identify studies wherein the incidence, risk factors and survival of gastric cancer with peritoneal metastases were investigated. Of all 38 potentially eligible studies, 17 studies were included based on the eligibility criteria. The incidence of synchronous gastric peritoneal metastases was reviewed for population-based studies (10–21%), for observational cohort studies (2–15%) and for surgical cohort studies (13–40%). Potential risk factors for synchronous gastric peritoneal metastases were younger age, non-cardia gastric cancer, female sex, signet ring cell carcinoma, diffuse type histology or linitis plastica, T4 stage, Hispanic ethnicity and more than one metastatic location. Synchronous peritoneal metastases are commonly diagnosed in patients with gastric cancer with an incidence up to 21% in recent population-based studies. Furthermore, prognosis of patients with gastric peritoneal metastases is poor with median overall survival ranging from 2 to 9 months. The high incidence and poor prognosis require intensive research on diagnostic features and effective treatment options to improve survival.

## 1. Introduction

Gastric cancer is one of the most common cancers worldwide with an incidence of over one million cases in 2020. It is the third most common cause of cancer-related death in the world, with almost 800,000 deaths a year [1]. Among Asian men, gastric cancer is even the most commonly diagnosed cancer and the leading cause of cancer death [2]. Due to the lack of early symptoms, patients with gastric cancer are often diagnosed in an advanced stage, which generally leads to a poor prognosis [3].

The peritoneal cavity is a well-known metastatic site in gastric cancer. For a long time, patients with isolated peritoneal metastases regardless of their origin had a dismal prognosis, and therapeutic options were scarce. However, several studies investigating the effect of cytoreductive surgery (CRS) and hyperthermic intraperitoneal chemotherapy (HIPEC) in patients with gastric, colorectal and ovarian peritoneal metastases have suggested an improvement in survival in carefully selected patients [4,5,6,7]. A randomised controlled trial (PERISCOPE II, NCT03348150) currently enrols gastric cancer patients with isolated limited peritoneal metastases to investigate whether CRS-HIPEC provides a survival benefit compared to systemic chemotherapy alone [8,9]. For patients with more extensive disease, new therapeutic options such as pressurized intraperitoneal aerosol chemotherapy (PIPAC) or normothermic intraperitoneal chemotherapy are being studied in clinical trials [10,11]. Awaiting the results of these trials, the current standard treatment in the Netherlands for this patient group remains palliative systemic chemotherapy, although the beneficial effect of current chemotherapeutic regimens is probably limited [12]. In patients with HER2-positive gastric cancer, the addition of trastuzumab may be considered as the randomised controlled ToGa-trial showed that this prolonged survival in advanced gastric cancer as compared to systemic chemotherapy alone [13].

The evolution and refinement of new techniques such as CRS-HIPEC, PIPAC and normothermic intraperitoneal chemotherapy have generated a renewed interest in the treatment of gastric peritoneal metastases. However, the burden of peritoneal metastases from gastric cancer is currently not well described. Detailed information on this topic will be helpful in counselling of patients and will guide future research directions. Especially, knowledge about risk factors for peritoneal metastases and the impact of survival may contribute to a tailored approach in treatment of patients with gastric cancer. The aim of this systematic review was to provide an overview of the incidence, risk factors and survival of gastric cancer with peritoneal metastases.

## 2. Materials and Methods

This systematic review was reported according to the Preferred Reporting Items for Systematic Reviews and Meta-Analyses (PRISMA) statement [14]. Two researchers (A.R. and R.J.L.) independently performed the literature search, study selection, data collection, risk of bias assessment and data synthesis. Inter-reviewer disagreements were resolved by achieving consensus between the two researchers.

### 2.1. Eligibility Criteria

Studies were considered potentially eligible if: (1) patients with gastric cancer were included and (2) the incidence, risk factors and/or survival of synchronous peritoneal metastases were analysed in the setting of a population-based or observational cohort. Furthermore, in a specific subgroup, studies were considered potentially eligible if patients who underwent diagnostic laparoscopy for staging of gastric cancer were investigated. Studies reporting on incidence were considered eligible if synchronous peritoneal metastases were reported as the proportion of all patients with gastric cancer. Studies reporting on risk factors were considered eligible if: (1) multivariable regression analyses were performed and (2) an odds ratio or relative risk were reported as outcome measure. Furthermore, studies reporting exclusively on patients with gastro-oesophageal junction cancer, case-reports, systematic reviews and studies with a publication year before 2000 were excluded. No language restrictions were applied.

### 2.2. Search Strategy

On 15 August 2021, PubMed/MEDLINE, EMBASE and Cochrane were systematically searched with a date restriction from 2000 to 2021. Full search queries are presented in Appendix A. The references of all eligible manuscripts were searched for additional eligible studies.

### 2.3. Study Selection

Titles and abstracts were screened for potentially eligible studies based on the predefined eligibility criteria. Afterwards, all potentially eligible studies were thoroughly read screened for final inclusion.

### 2.4. Data Collection

Data were collected by two researchers (A.R. and R.J.L.) using a standardised form that contained the following items: year of publication, study design, study setting, country, enrolment period, total number of patients, study population and the three outcomes under investigation (incidence, risk factors and survival).

### 2.5. Synthesis of Results

Results of all studies considered eligible were descriptively presented. Due to the high degree of heterogeneity across the included studies (i.e., study design, differences in study population), no meta-analysis was performed.

## 3. Results

### 3.1. Study Selection

After title and abstract screening, 38 studies were considered potentially eligible. After full text screening, seventeen studies were included [12,15,16,17,18,19,20,21,22,23,24,25,26,27,28,29,30]. The study flowchart and reasons for exclusion are shown in Figure 1 and Appendix B. In sixteen studies, information on incidence numbers of synchronous gastric peritoneal metastases was provided [12,16,17,18,19,20,21,22,23,24,25,26,27,28,29,30]. Risk factors for gastric peritoneal metastases were reported in four studies [15,16,17,18]. Survival was also reported in four studies [12,15,16,28,29].

### 3.2. Study Characteristics

Of all included studies, five studies were population-based studies [12,15,16,17,18], six studies were observational cohort studies [19,20,21,22,23,24] and six studies reported surgical cohorts of patients who underwent a staging laparoscopy [25,26,27,28,29,30]. Study characteristics and outcome measures of all studies are presented in Table 1A,B and Table 2. The five population-based studies were published between 2013 and 2021 [12,15,16,17,18]. The number of included patients ranged from 5220 to 34,943 (Table 1A). The six observational cohort studies were published between 2003 and 2015 and the number of included patients ranged from 1108 to 4559 (Table 1B) [19,20,21,22,23,24]. The six studies that reported the incidence of gastric peritoneal metastases of patients who underwent staging laparoscopy were published between 2013 and 2020, and the number of included patients ranged from 89 to 867 (Table 2) [25,26,27,28,29,30].

### 3.3. Incidence

#### 3.3.1. Population-Based Studies

Incidence of synchronous gastric peritoneal metastases was reported in five population-based studies from Sweden [17], the United States [18] and the Netherlands [12,15,16]. The proportions of patients presenting with peritoneal metastases from gastric cancer ranged from 10% to 21%.

#### 3.3.2. Observational Cohort Studies

Incidence of synchronous gastric peritoneal metastases was reported in six observational cohort studies from Germany [19], South-Korea [20,22], Japan [21,24], China [21] and the United States [23]. The proportion of patients with gastric peritoneal metastases ranged from 2% to 15%.

#### 3.3.3. Surgical Cohort Studies

Six studies reported the incidence of synchronous gastric peritoneal metastases of patients who underwent a staging laparoscopy. Patient in these studies were eligible for curative intent surgery and no systemic metastases after radiological staging [25,26,27,28,29,30]. The studies in this subgroup were conducted in the United States [25], China [26,27], Pakistan [28] and the United Kingdom [29,30]. The reported incidence ranged from 13% to 40%.

### 3.4. Risk Factors

Risk factors for synchronous gastric peritoneal metastases were reported in four studies [15,16,17,18]. Younger age [15,16,17], non-cardia cancer [15,16,17], female sex [15,16,17], signet ring cell carcinoma [16,17], diffuse type histology or linitis plastica [15,16], T4 stage [16], Hispanic ethnicity [18] and more than one location of metastases [15] were associated with an increased risk of gastric peritoneal metastases. Contradicting results were published regarding the association with positive lymph node status [15,16]. Details on risk factors are presented in Table 3.

### 3.5. Survival

Survival was reported in three population-based studies and in one surgical cohort study [12,15,16,29]. One study reported a median overall survival (OS) of 4.0 months in patients with gastric peritoneal metastases [16]. Another study reported survival of gastric peritoneal metastases by histological subtype with a median OS of 4.6 months for diffuse type gastric cancer versus 5.1 months for intestinal type gastric cancer with peritoneal metastases [15]. Furthermore, a study on gastric peritoneal metastases reported a median OS of 2.1 months in patients who did not receive systemic therapy versus 9.4 months in patients who received systemic therapy [12]. A study documented a median OS of 7 months in a gastric cancer patients cohort that underwent staging laparoscopy [29].

## 4. Discussion

In this systematic review, the proportion of patients with synchronous peritoneal metastases from gastric cancer origin ranged from 10–21% in population-based studies [12,16,17,18], from 2–15% in observational cohort studies [19,20,21,22,23,24] and from 13–40% in surgical cohort studies [25,26,27,28,29,30]. Interestingly, the highest incidence of synchronous peritoneal metastases (21%) was reported in the most recent population-based study [12]. This may be attributable to the improvement of imaging techniques resulting in a higher detection rate of the typically small peritoneal lesions as well as a higher awareness towards peritoneal metastases amongst radiologists. Moreover, the introduction of a standard diagnostic laparoscopy in the staging guidelines of operable patients with resectable gastric cancer will have contributed to the increased documentation of peritoneal metastases. Identified risk factors for gastric peritoneal metastases were younger age, non-cardia cancer, female sex, signet ring cell carcinoma, diffuse type histology or linitis plastica, T4 stage and Hispanic ethnicity [15,16,17,18]. Median OS in patients with gastric peritoneal metastases ranged from 2 to 9 months [12,15,16,29].

To the best of our knowledge, this is the first comprehensive systematic review providing an overview on incidence, risk factors and survival for synchronous gastric peritoneal metastases. Previous studies have performed a systematic review on gastric cancer in general but reported very limited information about peritoneal metastases with none of these studies focusing specifically on the incidence of synchronous peritoneal metastases [31,32]. From these studies, it can be concluded that the proportion of patients presenting with metastases at any location increased over time from 24% in 1990 to 44% in 2011. The peritoneum is recognized as one of the most common metastatic sites in gastric cancer patients, ranking second after the liver [3,17]. Again, improved radiologic and staging techniques probably explain the stage migration towards more patients with metastatic disease. One review on gastric cancer confirmed the striking difference of a much higher incidence of gastric cancer in Asian countries than in Western countries, as well as a less advanced stage at the time of diagnosis [33]. The latter may be explained by the mass screening programs for gastric cancer in high-incidence regions such as Japan and Korea, aiming to diagnose the cancer at an early stage [34].

In the current review, several risk factors to develop peritoneal metastases were identified, among which are a younger age. Interestingly, a meta-analysis on young patients with gastric cancer also reported that these patients were more often females with diffuse gastric cancer and signet ring cell carcinoma and were more often diagnosed with peritoneal metastases [35]. Therefore, young patients may have a poorer tumour biology and subsequently may be more at risk for peritoneal metastases. On the other hand, younger patients are usually in a good condition and are thus more likely to receive a thorough diagnostic work-up which increases the chance of discovering peritoneal metastases. Therefore, it remains unknown whether the higher incidence of peritoneal metastases in younger patients reflects a more aggressive tumour biology or whether this finding is biased by an intensified diagnostic workup.

Other risk factors, such as a T4 stage and signet ring cell differentiation, were previously identified to be associated with an increased incidence of peritoneal metastases from colorectal cancer [36]. Furthermore, linitis plastica, tumour-positive lymph nodes and a primary tumour not located in the cardia were previously reported as risk factors for metastases in gastric cancer patients [37]. This highlights the role of a more advanced tumour stage in the development of peritoneal metastases. Remarkably, in one study, tumour-positive lymph nodes were associated with a higher rate of systemic metastases but with a lower risk of peritoneal metastases [15]. At first, this may seem contradictory, but this can be explained by the fact that this study was performed in patients presenting with metastatic disease only. Patients with lymph node involvement and systemic metastases on computed tomography (CT) probably have not undergone a staging laparoscopy since they are already considered to have unresectable disease. As a result, peritoneal metastases may have been missed in many patients as they are usually hard to diagnose by radiologic imaging alone.

Staging laparoscopy is frequently carried out in patients with (advanced) gastric cancer eligible for curative intent surgery and without metastases after radiology staging. In this systematic review, studies on patients who underwent staging laparoscopy generally reported a higher incidence of gastric peritoneal metastases compared to the other studies, up to 40%. This proportion is comparable to the numbers of a recent review specifically on gastric cancer patients undergoing staging laparoscopy [38]. This emphasizes the importance of a staging laparoscopy in patients with gastric cancer. Less invasive diagnostic modalities, such as (positron emission tomography) CT and magnetic resonance imaging, need to be further improved to increase their accuracy for diagnosing peritoneal metastases.

As shown in this review, survival of gastric peritoneal metastases is poor, ranging from 2 to 9 months, depending on systemic therapy or histological subtype. Similar poor survival outcomes for patients receiving best supportive care, or systemic therapy only, were previously reported for peritoneal metastases of other primary tumours, such as colorectal and pancreatic cancer, which emphasizes the need for new treatment options for patients with peritoneal metastases, regardless of the origin of the tumour [39,40,41]. In gastric cancer with peritoneal metastases, experimental treatment options such as CRS-HIPEC or PIPAC are currently being investigated [8,10]. Although limited literature is available about this experimental treatment, preliminary results seem promising [6,7,42]. Furthermore, it needs to be investigated whether patients with peritoneal metastases may also benefit from new systemic treatment strategies such as docetaxel-based triplet FLOT therapy (fluorouracil plus leucovorin, oxaliplatin and docetaxel) that has been shown to improve survival in patients with locally advanced resectable gastric cancer [43].

This review has several limitations. Firstly, some population-based studies used the same data registries which results in overlapping use of patient characteristics [12,15]. However, these studies reported on different outcomes and therefore did not result in duplication of data. Secondly, the population-based studies were performed in western countries, whereas the observational cohort studies were mostly performed in Asian countries. The prevalence of gastric cancer in western countries is low compared to the prevalence in Asian countries, and types of histology vary among these different parts of the world where diffuse type is more common in Asian countries [1,2,43,44,45]. The observational cohort studies revealed large heterogeneity within and across these studies. This may lead to an incomplete overview of patients diagnosed with synchronous gastric peritoneal metastases in non-western countries. Finally, this systematic review focused on synchronous peritoneal metastases only, whereas it is known that metachronous peritoneal metastases frequently occur after curative treatment for gastric cancer. Recent literature showed that the peritoneum (36%) was the most common initial site of recurrence after potentially curative gastric cancer surgery [46]. Population-based studies with adequate follow-up to include metachronous peritoneal metastases are therefore designated to provide a more accurate overview of the total burden of peritoneal metastases from gastric cancer.

To conclude, in this systematic review, synchronous peritoneal metastases were frequently commonly diagnosed in patients with gastric cancer with an incidence up to 21% in most recent population-based studies. Furthermore, prognosis of patients with gastric peritoneal metastases is poor. Given the high incidence and poor prognosis, this patient category is an important focus for future research on diagnostic features and effective treatment options to improve survival.

## Figures and Tables

**Figure 1 jcm-10-04882-f001:**
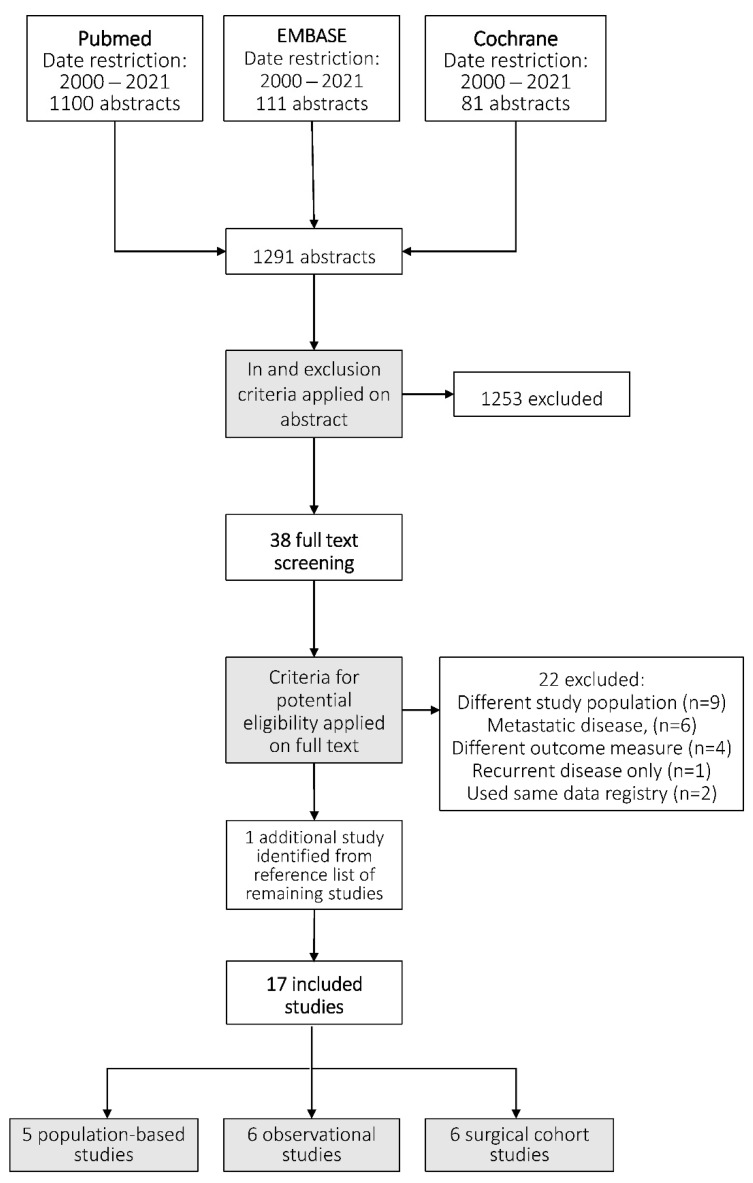
Literature search and study selection. Details of the literature search and study selection are presented in the Appendix A and Appendix B, respectively.

**Table 1 jcm-10-04882-t001:** Study characteristics of (**A**) population-based studies and (**B**) observational cohort studies.

**(A)**									
	**Design**				**Outcomes**				
**First Author** **Year**	**Enrolment Period**	**Country**	**Setting**	**Total Patients (n)**	**Incidence of GPM**				**Risk Factors for GPM**
					Reported	*n*	%	Reported	Risk factors
Koemans [12] ^a^ 2021	2008–2017	The Netherlands	Nationwide register	12,504	Yes	2607	21%	No	
Koemans [15] ^a^ 2020	1999–2017	The Netherlands	Nationwide register	34,943	No			Yes	− Non-cardia cancer − Age <45 years − Female sex − T2–T4 stage − >1 metastasis location − Diffuse type histology according to Lauren classification − Diagnosis in 2013–2017
Thomassen [16] 2013	1995–2012	The Netherlands	Regional register of southern part of the Netherlands	5220	Yes	706	14%	Yes	− Signet ring cell carcinoma − Linitis plastica − Non-cardia cancer − Age <60 years − Female sex − T3/T4 stage − N1–N3 stage − Poor tumour differentiation
Riihimäki [17] 2016	2002–2012	Sweden	Nationwide register	7559	Yes	936	12%	Yes	− Signet ring cell carcinoma − Non-cardia cancer − Age <60 years − Female sex
Choi [18] 2020	2004–2014	United States	State register of California	16,275	Yes	1691	10%	Yes	− Hispanic ethnicity
**(B)**									
	**Design**				**Outcomes**				
**First Author** **Year**	**Enrolment Period**	**Country**	**Setting**	**Total Patients (*n*)**	**Incidence of GPM**				
					Reported		n		%
Seyfried [19] 2015	1986–2013	Germany	Single centre	1108	Yes		158		14%
Park [20] 2010	2000–2005	Korea	Single centre	3193	Yes		104		3%
Yu [21] 2010	1993–2006 1980–2003	JapanChina	Multicentre	2063 2496	Yes		42 324		2% 13%
Kim [22] 2003	1988–1998	Korea	Single centre	1833	Yes		267		15%
Yao [23] 2005	1985–1999	United States	Single centre	1897	Yes		200		11%
Isobe [24] 2013	1977–2006	Japan	Single centre	3818	Yes		447		12%

GPM indicates gastric peritoneal metastases. ^a^ Study contains same data register from The Netherlands.

**Table 2 jcm-10-04882-t002:** Study characteristics of studies on gastric cancer and staging laparoscopy.

	Design					Outcomes		
First Author Year	Enrolment Period	Country	Setting	Inclusion Criteria	Total Patients (*n*)	Incidence of GPM		
						Reported	n	%
Allen [25] 2020	1995–2018	United States	Single centre	− Gastric cancer − Histopathological confirmation by endoscopic ultrasound − Underwent SL	867	Yes	175	20%
Hu [26] 2016	2004–2014	China	Single centre	− Advanced stage gastric cancer − Underwent SL	582	Yes	118	20%
Yang [27] 2020	2014–2019	China	Single centre	− Gastric cancer − Potentially resectable − Histopathological confirmation − M0 on preoperative screening − Underwent SL	672	Yes	89	13%
Bhatti [28] 2014	2005–2012	Pakistan	Single centre	− Gastric cancer − Potentially resectable − M0 on preoperative screening − Underwent SL	89	Yes	36	40%
Convie [29] 2015	2007–2013	United Kingdom	Single centre	− Gastric cancer − Potentially resectable − M0 on preoperative screening − Underwent SL	159	Yes	36	23%
Munasinghe [30] 2013	2006–2010	United Kingdom	Single centre	− Gastric cancer − Potentially resectable − Underwent SL	142	Yes	19	13%

GPM indicates gastric peritoneal metastases. SL indicates staging laparoscopy.

**Table 3 jcm-10-04882-t003:** Risk factors for synchronous gastric peritoneal metastases.

**Study**	**Groups**	**OR**	**95% CI**
**Age**			
Koemans et al. (2020) [15]	<45 years	Ref.	Ref.
	46–60 years	0.74	0.6–0.9
	61–75 years	0.62	0.5–0.8
	>75 years	0.52	0.4–0.7
Thomassen et al. (2013) [16]	<60 years	Ref.	Ref.
	60–69 years	0.7	0.5–0.9
	70–79 years	0.5	0.4–0.6
	>80 years	0.3	0.2–0.3
Riihimaki et al. (2015) [17]			
Single metastasis	<60 years	Ref.	Ref.
	70–79 years	0.5	0.4–0.7
	>79 years	0.3	0.2–0.4
Multiple metastases	<60 years	Ref.	Ref.
	60–69 years	0.8	0.7–1.0
	70–79 years	0.5	0.4–0.6
	>79 years	0.2	0.2–0.3
**Study**	**Groups**	**OR**	**95% CI**
**Location of primary gastric tumour**			
Koemans et al. (2020) [15]	OGJ/cardia	Ref.	Ref.
	Proximal/Middle stomach	2.4	2.1–2.8
	Distal stomach	2.7	2.3–3.1
	Overlapping location	3.6	3.1–4.1
Thomassen et al. (2013) [16]	Non-cardia	Ref.	Ref.
	Cardia	0.4	0.3–0.5
	Overlapping lesions/NOS	1.3	1.0–1.6
Riihimaki et al. (2015) [17]			
Single metastasis	Cardia	Ref.	Ref.
	Fundus/Corpus	1.7	1.3–2.2
	Antrum/Pylorus	1.8	1.3–2.3
Multiple metastases	Cardia	Ref.	Ref.
	Fundus/Corpus	1.8	1.4–2.2
	Antrum/Pylorus	1.6	1.2–2.0
**Sex**			
Koemans et al. (2020) [15]	Male	Ref.	Ref.
	Female	1.5	1.3–1.6
Thomassen et al. (2013) [16]	Male	Ref.	Ref.
	Female	1.2	1.0–1.5
Riihimaki et al. (2015) [17]			
Single metastasis	Male	Ref.	Ref.
	Female	1.1	1.0–1.4
Multiple metastases	Male	Ref.	Ref.
	Female	1.3	1.1–1.5
**Histology**			
Koemans et al. (2020) [15]	Intestinal type	Ref.	Ref.
	Diffuse type	2.8	2.5–3.1
	Mixed type	2.1	1.7–2.7
Thomassen et al. (2013) [16]	Adenocarcinoma	Ref.	Ref.
	Signet ring cell carcinoma	1.7	1.4–2.2
	Linitis plastica	2.0	1.5–2.8
Riihimaki et al. (2015) [17]			
Single metastases	Adenocarcinoma	Ref.	Ref.
	Signet ring cell carcinoma	2.5	2.0–3.1
Multiple metastases	Adenocarcinoma	Ref.	Ref.
	Signet ring cell carcinoma	2.3	1.9–2.7
**Study**	**Groups**	**OR**	**95% CI**
**T stage**			
Koemans et al. (2020) [15]	T1	Ref.	Ref.
	T2–3	2.1	1.3–3.2
	T4	3.0	1.9–4.7
Thomassen et al. (2013) [16]	T1–2	Ref.	Ref.
	T3	2.4	1.7–3.3
	T4	2.9	2.1–4.0
**N stage**			
Koemans et al. (2020) [15]	N0	Ref.	Ref.
	N1–2	0.4	0.3–0.4
	N3	0.3	0.2–0.3
Thomassen et al. (2013) [16]	N0	Ref.	Ref.
	N+	4.0	2.2–7.3
**Ethnicity**			
Choi et al. (2020) [18]			
Non-Hispanic white vs. Hispanic	Non-Hispanic white	Ref.	Ref.
	Hispanic	1.9	1.6–2.1
Asian/other vs. Hispanic	Asian/other	Ref.	Ref.
	Hispanic	1.5	1.3–1.7
**Number of metastatic locations**			
Koemans et al. (2020) [15]	1 metastasis	Ref.	Ref.
	>1 metastases	1.6	1.5–1.8

Non-significant risk factors were excluded. OR indicates odds ratio. CI indicates confidence interval. Ref. indicates the reference category.

## Data Availability

Not applicable.

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
