# Peer review of "The Burden of Peritoneal Metastases from Gastric Cancer: A Systematic Review on the Incidence, Risk Factors and Survival"

_jcm, 2021, doi:10.3390/jcm10214882_

Round 1

Reviewer 1 Report

In the current manuscript the authors provided an overview about the incidence, risk factors and survival of synchronous peritoneal metastases from gastric cancer. In particular, authors performed a literature search, selecting all studies ranging from 2000 to 2021, using Pubmed, EMBASE and Cochrane databases and, after the application of the eligibility criteria, they included only 17 studies in the analysis. These studies are divided in tree groups: population-based studies, observational cohort studies and surgical cohort studies. The reported incidence of synchronous peritoneal metastases is different among these studies with 10-21% for population-based studies, 2-15% for observational cohort studies and 13-40% for surgical cohort studies. In addition, from the performed analysis, the authors reported several risk factors for synchronous peritoneal metastases such as female sex, younger age, ethnicity and histological type. Finally, the authors investigated the survival rate of patients with gastric peritoneal metastasis with median overall survival ranging from 2 to 9 months.

Strengths: As gastric cancer and the associated metastases represents one of the most diffuse malignancy in the world, knowing its incidence and its poor prognosis can be an interesting focus for the diagnostic research. This manuscript represents a good tool for clinicians to have updates about gastric peritoneal metastases incidence and prognosis.

Minor revision

  • Authors should correct page number
  • The abstract conclusion is the same of the discussion conclusion “Given the high incidence and poor prognosis, this patient category is an important focus for future research on diagnostic features and effective treatment options to improve survival”. The authors should change it
  • In the section “Study characteristics” the authors said: The six studies that reported the incidence of gastric peritoneal metastases of patients who underwent staging laparoscopy were published between 2014 and 2020… in the table 2 it is indicated that Munasinghe et alis 2013 publication. So, authors should change the sentence.
  • In the figure 1 authors should correct English mistake “38 full tekst screening”
  • Authors should follow the authors guideline for the references. In the discussion section some references are not suitable with guidelines: “metastases from gastric cancer origin ranged from 10%-21% in population-based stud-ies[{Koemans, 2021 #5}10, 14-16]” … “associated with an increased incidence of peritoneal metastases from colorectal cancer34.”

Authors should fix author contributions section

Reviewer 2 Report

As the authors point out,  this review only captures studies of synchronous periotoneal mets but not studies which include metachronous peritoneal mets, therefore the extent that prognosis in patients with peritoneal mets can predicted is limited in this study.  Peritoneal limited disease in GC is heterogenous and prognosis is dependent on tumour biology (PCI, extent of ascites as seen in  the Phoenix Trial JCO 2018 Ishigami), interventions used and patient selection. Indeed in Phoenix the median survival was 15.2m even with standard chemo and the French Gastro-chip study shown a proportion of patients who were disease free at 5yr after CRS HIPEC for low PCI GC. Peritoneal disease is an impt area of therapeutic interest in GC, perhaps the authors could reference studies such as normothermic IP chemo in addition to HIPEC and PIPAC.  

Secondly in the discussion, the authors write "The prevalence of gastric cancer in Western countries is low compared to the prevalence in Asian countries and types of histology vary among these different parts of the world where intestinal type is more common in Asian countries[1, 2, 42]."  But diffuse GC is reportedly more common in Asian countries c/w Western countries eg. frequency seen in  RESOLVE study, ARTIST-2 studies vs FLOT4 study
